# Engineering of high-precision base editors for site-specific single nucleotide replacement

Junjie Tan [1], Fei Zhang[1,2], Daniel Karcher[1] & Ralph Bock [1]

RNA-guided nucleases of the CRISPR/Cas type can be repurposed as programmable nucleotide deaminases to mediate targeted nucleotide substitutions. Such base editors have enormous potential in genome editing, gene therapy and precision breeding. However, current editors suffer from limited specificity in that they edit different and/or multiple bases within a larger sequence window. Using cytidine deaminase base editors that elicit C-to-T mutations, we show here that high editing precision can be achieved by engineering the connection between the deaminase domain and the Cas domain of the editor. By systematically testing different linker sequences and removing non-essential sequences from the deaminase, we obtain high-precision base editors with narrow activity windows that can selectively edit a single cytidine at a specific position with high accuracy and efficiency. These base editors will enable the use of genome editing in applications where single-nucleotide changes are required and off-target editing of adjacent nucleotides is not tolerable.

---

[1] Max-Planck-Institut für Molekulare Pflanzenphysiologie, Am Mühlenberg 1, 14476 Potsdam-Golm, Germany. [2]Present address: National Key Laboratory of Crop Genetic Improvement, Huazhong Agricultural University, 430070 Wuhan, Hubei, China. Correspondence and requests for materials should be addressed to R.B. (email: rbock@mpimp-golm.mpg.de)

CRISPR–Cas systems represent an adaptive immune system in bacteria that promotes antiviral defense[1,2]. Several such systems, especially the one based on the Cas9 enzyme from *Streptococcus pyogenes* (SpCas9), have been successfully repurposed for genome editing in a wide range of organisms[3–8]. Cas9 is an endonuclease with two nuclease domains, each cleaving one strand of the target DNA[9,10]. Upon repair of the double-strand break, deletions (or insertions) can occur that inactivate the target gene[11]. Although this method provides a highly efficient tool in functional genomics and is also suitable to reach a limited number of breeding goals by knocking out genes for unwanted traits in crops[12,13], more precise DNA editing tools are needed for all applications requiring introduction of specific base changes into target genes, such as precision breeding and gene therapy. Most hereditary diseases in humans involve single-point mutations, the correction of which will require extraordinary accuracy of site-specific editing, ideally without any off-target effects[14,15]. Recently, base editors have been developed that convert Cas endonucleases into programmable nucleotide deaminases[16–18], thus facilitating the introduction of C-to-T mutations (by C-to-U deamination) or A-to-G mutations (by A-to-I deamination) without induction of a double-strand break[19,20]. Base editors comprise a nickase form of SpCas9 (nSpCas9, to stimulate cellular DNA mismatch repair) fused to a nucleobase deaminase enzyme as well as an inhibitor of base excision repair such as uracil glycosylase inhibitor (UGI).

The current severe limitation in the applicability of base editors lies in their low site selectivity. For example, C-to-T base editors can potentially edit any C that resides in an approximately 4–5 nt (in some systems up to 9 nt) wide window within the protospacer[16,17,21]. However, some human disease-associated alleles such as the Alzheimer's disease-associated gene *APOE4* and the β-thalassemia locus *HBB* have multiple Cs around the targeted C within the activity window, and the editing of additional Cs can potentially cause deleterious effects[16,22]. Therefore, efforts have been made to reduce the width of the editing window, and introduction of mutations that reduce the deaminase activity were shown to have some positive effects[23–25]. However, in addition to the undesirable reduction of the editing activity, the beneficial effect of these mutations on editing specificity was dependent on the sequence context[23,24].

Here, we have attempted to provide a more general solution to the specificity problem of base editors. By engineering the linker sequences and eliminating non-essential sequences, we obtain high-precision base editors with narrow activity windows that are capable of selectively editing a single cytidine residue with high accuracy and efficiency. Our improved base editors will likely facilitate applications in genome editing, gene therapy, and precision breeding.

## Results

### Rigid linkers improve precision of APOBEC1-based editors.

We hypothesized that the positioning on the target sequence of the Cas9 protein relative to the deaminase domain (i.e., their physical distance) and the rigidity of the connection between these two domains of the base editor determine the width of the editing window, and hence the precision of the base editor. In previous studies, a 16 amino acid (aa) flexible linker (XTEN) has been identified as the best compromise between editing efficiency and specificity[16]. Using L-canavanine selection in yeast[17], we first investigated the effects of length and rigidity of the linker between APOBEC1 and nCas9 (Cas9 nickase) on base editing precision and efficiency when targeting several sites in the *Can1* gene (Fig. 1; Supplementary Figure 1) that contain Cs within the activity window of the base editor BE3 (ref. [16]). L-Canavanine is a

highly toxic analog of the proteinogenic amino acid arginine, and mutations inactivating the uptake protein Can1 confer resistance to canavanine. We used an inducible base editor construct, determined the optimal induction time, and then tested 10 different rigid linker sequences (containing the amino acid proline that, due to its secondary amine, confers conformational rigidity) in comparison to the commonly used XTEN flexible linker (Supplementary Figures 1 and 2). Consistent with previous reports[16], the base editor BE3 (containing the XTEN linker) allowed editing at all Cs within a window of nine nucleotides (Fig. 1; Supplementary Figure 3). Omission of the linker sequence or use of a very short rigid linker (i.e., the 3 aa linker PAP) abolished editing nearly completely. Interestingly, rigid linkers of 5–7 aa made editing substantially more precise, with the seven aa linker PAPAPAP largely restricting editing to positions −15 and −16 (Fig. 1). Longer linkers resulted in reduced editing accuracy, suggesting that a seven aa rigid linker is optimal.

It was reported that mutations in the APOBEC1 domain of BE3 can also narrow the base editing width. We, therefore, compared the base editing outcome of BE3, YEE-BE3 (the optimal BE3 variant[23]), and BE-PAPAPAP when targeting the *Can1* sites. We found that YEE-BE3, although mainly editing $C_{-15}$ or $C_{-16}$, suffered from strongly reduced editing activity at these sites (Supplementary Figure 4). Although it will be important to confirm this deficit for additional sequence contexts, this finding is consistent with a recent study that also reported low editing efficiency of the YEE-BE3 base editor[24].

Previous work has mostly investigated the activity of base editors in favorable sequence contexts, with relatively few C targets within the protospacer sequence. To develop a more rigorous (and *Can1*-independent) assay for base editor specificity, we also investigated the worst-case scenario, in which all nucleotides within the BE3 activity window are Cs (i.e., a nonacytidine motif from −13 to −21). Analysis of editing products by deep sequencing revealed that base editors with 5–7 aa rigid linkers mainly edited at positions $C_{-14}$ to $C_{-16}$. These editors showed greatly improved site selectivity and a narrowed editing window, while retaining up to 90% of the editing efficiency of the original BE3 (Supplementary Figure 5a, b). Importantly, when editing product distribution was analyzed, BE3-treated sequences mostly contained four simultaneously edited bases, whereas short rigid linker-containing base editors predominantly generate products with one to three edited bases (Supplementary Figure 5c), thus providing further evidence for short rigid linkers leading to more precise editing.

### Engineering of improved CDA1-based editors.

To test whether other base editors can also be improved by engineering the linker region connecting the nucleoside deaminase domain with the nCas9 domain, we next applied a similar strategy to CDA1, the AID homolog of sea lamprey[17] that has been reported to exhibit superior performance to APOBEC1 in certain sequence contexts[26]. When fused to nCas9 with flexible linkers up to 100 aa long[17], CDA1 conducts C-to-T conversion in a window of approximately −16 to −19. To better understand what influences the width of the activity window, we generated four constructs for direct comparison of N- and C-terminal fusions of APOBEC1 and CDA1 to nCas9, initially using the XTEN linker (Fig. 2a). When the APOBEC1 domain was fused to the C-terminus of nCas9 (cBE3), the editing activity was very low (Fig. 2b, c), consistent with previous observations[16]. By contrast, when CDA1 was fused to either the N-terminus or the C-terminus of nCas9, both fusions exhibited high editing efficiency. However, there was a remarkable difference in the width of the editing window, in that the N-terminal CDA1 (nCDA1-BE3) triggered editing in a

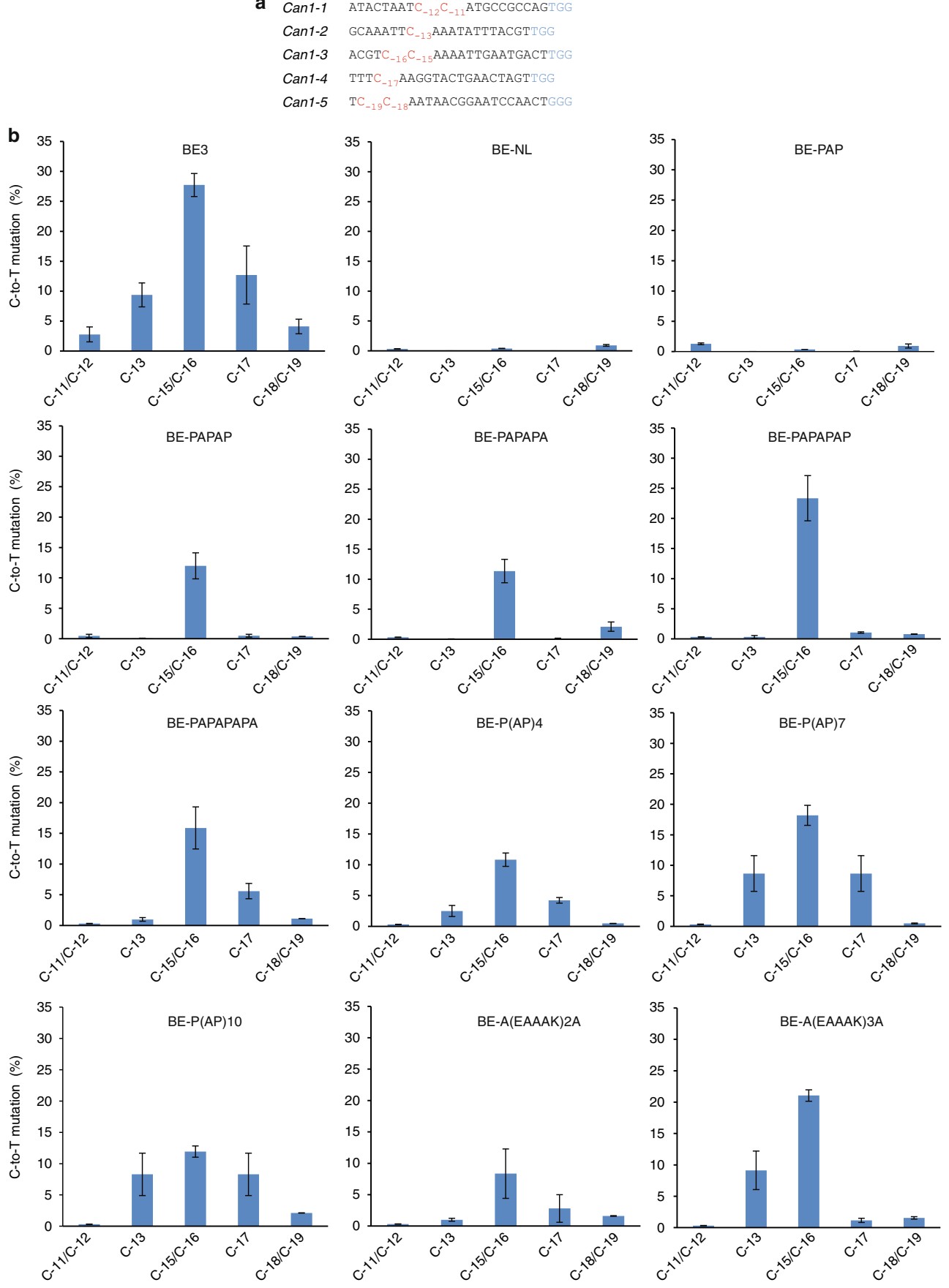

**Fig. 1** Rigid linkers narrow the width of the editing window of BE3. **a** Protospacers and PAM (blue) sequences of the genomic loci tested, with the target Cs shown in red. Subscript numbers indicate the positions of the cytidines relative to the PAM. C-to-T editing at any of the indicated Cs inactivates the Can1 transporter and thus causes resistance to canavanine[17]. **b** Editing efficiency and specificity of the base editors tested as determined by canavanine selection. The x-axis represents the target Cs within the protospacers. The y-axis shows their C-to-T editing frequency (see Methods and Supplementary Figure 1). Sequences of canavanine-resistant mutants aligned with the corresponding reference sequences are shown in Supplementary Figure 3. Values and error bars represent the mean and standard deviation of three independent biological replicates. Source data are provided as a Source Data file

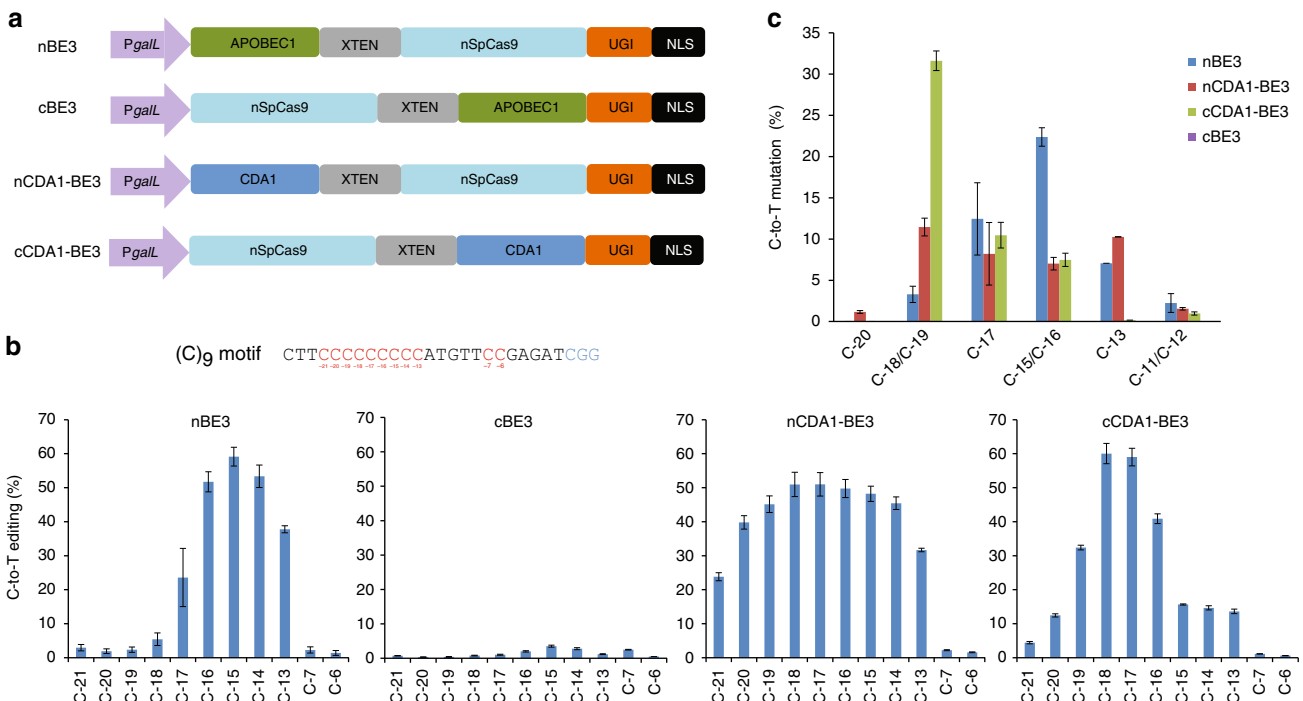

**Fig. 2** Comparison of N- and C-terminal deaminase fusions to nCas9. **a** Structure of nBE3 (=BE3; ref. [16]), cBE3, nCDA1-BE3, and cCDA1-BE3 driven by the *GalL* inducible promoter. In all constructs, the XTEN linker separates the nucleoside deaminase domain from the nCas9 domain. nSpCas9: *Streptococcus pyogenes* Cas9 nickase. **b** Base editors with the deaminase at the N-terminus show broadened base editing windows. The sequence of the target $(C)_9$ motif is shown with the numbers representing the position of possible editing targets (red) relative to the PAM (blue). % of C-to-T editing represents the percentage of total sequencing reads with the target C converted to T. **c** Base editing outcome of nBE3, cBE3, nCDA1-BE3, and cCDA1-BE3 targeting several sites containing target Cs at different positions (indicated on the x-axis) in the *Can1* gene (cf. Supplementary Figure 1; Supplementary Table 4). Values and error bars represent the mean and standard deviation of three independent biological replicates. Source data underlying panels **b** and **c** are provided as a Source Data file

much broader window when tested on either an oligo(C) substrate or target sites in the *Can1* gene (Fig. 2b, c; Supplementary Figure 6). The C-terminal fusion showed a more specific editing activity, peaking from $C_{-16}$ to $C_{-19}$, consistent with previous reports[17].

Comparative assessment of the specificity of previously generated base editors and our base editors on several genomic target sequences showed that, in many cases, some level of discrimination between adjacent Cs is possible, but the achievable precision depends on the sequence context and on the base editor used (Supplementary Figure 7). In general, the nCDA1-BE3 and cCDA1-BE3 editors display less dependence on the neighboring nucleotides and can edit target Cs efficiently even when located immediately after an A (e.g., *Can1–7* in Supplementary Figure 7), a context that is only very inefficiently edited by APOBEC1-based editors. Moreover, CDA1-based editors enhance product purity (Supplementary Figure 8), as reported previously[26].

In an attempt to further narrow the activity window of CDA1 editors, we removed the linker between CDA1 and Cas9, generating versions nCDA1-NL-BE3 and cCDA1-NL-BE3. Surprisingly, both linkerless fusions showed an unaltered activity window with largely unchanged editing efficiency at each C

within it (Supplementary Figure 9). This result suggests that the termini of CDA1 are inherently flexible and may act as linker-like sequences. We, therefore, tested the impact of N- and C-terminal truncations (removing potential linker-like fragments) on base editing.

A nuclear export signal (NES) was reported to reside in the C-terminus of the CDA1 homolog AID (ref. [27]), and its location corresponds to residues 199 to 208 in CDA1 (Fig. 3a). Deletion of the NES from AID increased the deamination efficiency of the enzyme[28–30]. We generated a series of 22 base editors with C-terminally truncated CDA1 versions fused to nCas9 (Fig. 3b) and tested them on two oligo(C) motifs (Fig. 4; Supplementary Figures 10 and 11). While removal of the NES had only small effects on editing efficiency and specificity (nCDA1Δ198-BE3), larger deletions made editing more precise and substantially narrowed the activity window of the base editors (Fig. 4). The enzyme tolerated truncations up to amino acid residue 158 without a significant loss in editing efficiency (Fig. 4; Supplementary Figures 10 and 11). The major gain in site selectivity was seen with the removal of at least 13–14 amino acids from the C-terminus of CDA1 (nCDA1Δ195-BE3, nCDA1Δ194-BE3; Fig. 4; Supplementary Table 1). Larger deletions had similar beneficial

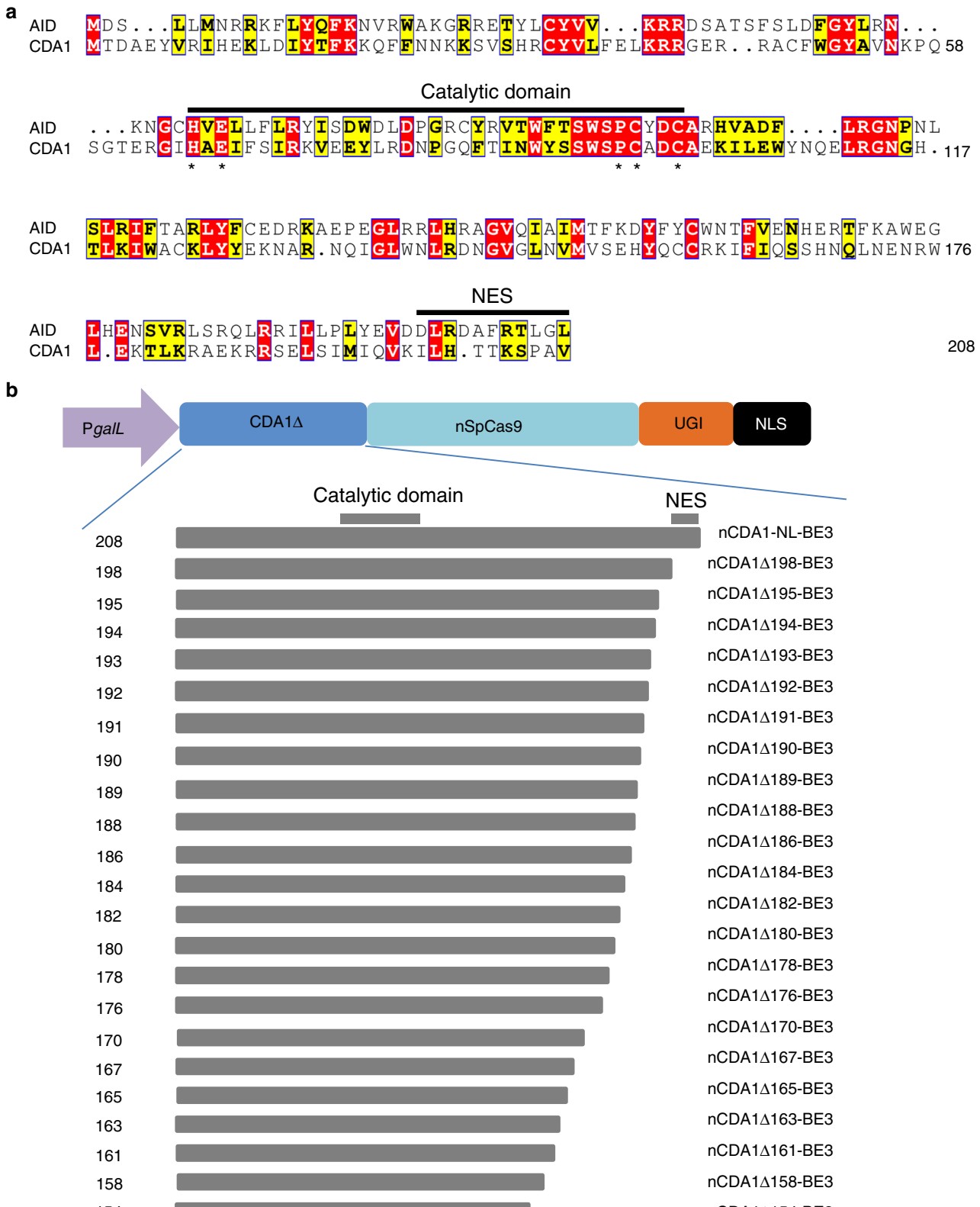

**Fig. 3** Design of base editors with truncated CDA1 domains. **a** Amino acid sequence alignment of CDA1 and human AID. The catalytic domain HxE-PCxxC and the nuclear export signal (NES) are indicated by black horizontal lines. The alignment was created by CLUSTALW (ref. [38]; https://www.genome.jp/tools-bin/clustalw) and graphically formatted with the help of the ESPript 3.0 server[39] (http://espript.ibcp.fr/ESPript/ESPript/). Identical amino acid residues are shaded in red, similar residues in yellow. **b** Schematic representation of base editors with C-terminal CDA1 truncations (named after the last CDA1 residue included)

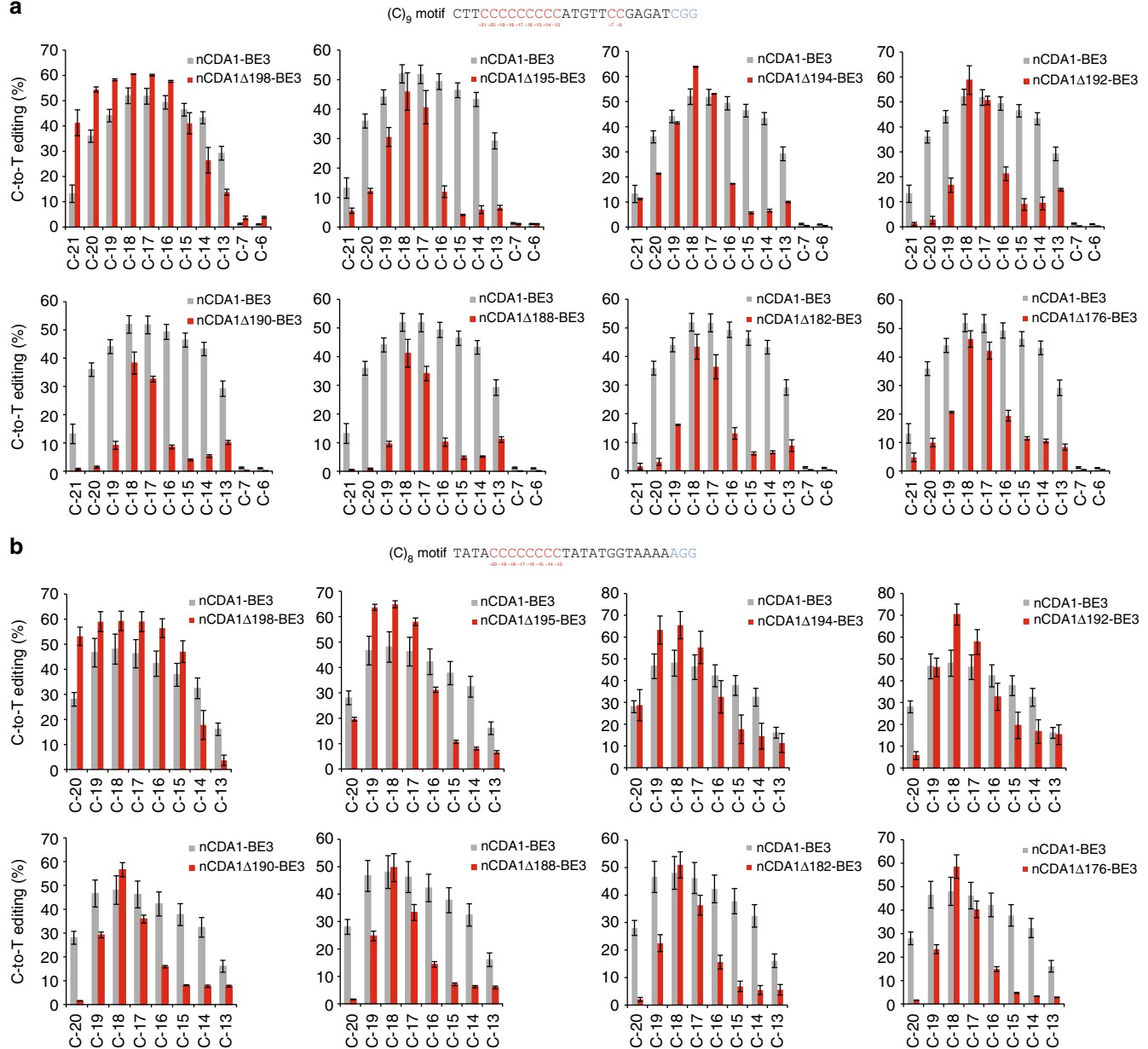

**Fig. 4** Effects of C-terminal truncations of the CDA1 domain on the width of the editing window of nCDA1-BE3 base editors. All base editor variants were tested on both $(C)_8$ (**a**) and $(C)_9$ (**b**) motifs (see Methods). Cs within each target region are shown in red, with the number below indicating their distance from the PAM (blue). The C-to-T conversion efficiencies are plotted for all Cs within the protospacer, and shown in comparison to the nCDA1-BE3 base editor with the full-length CDA1 (gray bars). Values and error bars represent the mean and standard deviation of three biological replicates. Source data are provided as a Source Data file. For a comparison with additional deletion constructs, see Supplementary Figures 10 and 11

effects on editing precision, although some of them displayed slightly reduced overall editing efficiency (Fig. 4; Supplementary Figures 10 and 11). Unlike the full-length base editor, the best-performing truncated variants showed a clear preference for one or two Cs within the oligo(C) stretch (e.g., nCDA1Δ194-BE3 for $C_{-18}$ and, to a lesser extent, $C_{-17}$ within the $(C)_9$ motif: Fig. 4a; nCDA1Δ192-BE3 and nCDA1Δ190-BE3 for $C_{-18}$ in the $(C)_8$ motif: Fig. 4b). By contrast, truncations at the N-terminus of CDA1 in cCDA1-BE3 had no significant effect on the width of the editing window (Supplementary Figure 12).

Tests on oligo(C) motifs represent the most stringent assays for site selectivity of base editors. However, such long C stretches would only rarely be targets of genome editing with base editors in vivo. To assess whether base editors with C-terminally

truncated CDA1 domains also show superior performance in more natural (heteropolymeric) genomic sequence contexts, we targeted four sites in the *Can1* gene, each of which contains at least one additional C directly adjacent or close to position $C_{-18}$. When the base editing outcome of nCDA1-BE3, cCDA1-BE3 and our base editors with truncated CDA1 domains were compared, our base editors displayed editing with much higher precision (Fig. 5). For all four tested sites, our base editors mainly edited position $C_{-18}$, with a 2- to 20-fold higher efficiency than other adjacent Cs (Fig. 5a). Importantly, the base editors also produced predominantly single-C-modified products at position $C_{-18}$ (accounting for 50–94% of all edited products), whereas nCDA1-BE3 and cCDA1-BE3 produced mainly double or triple modified products (Fig. 5b, c).

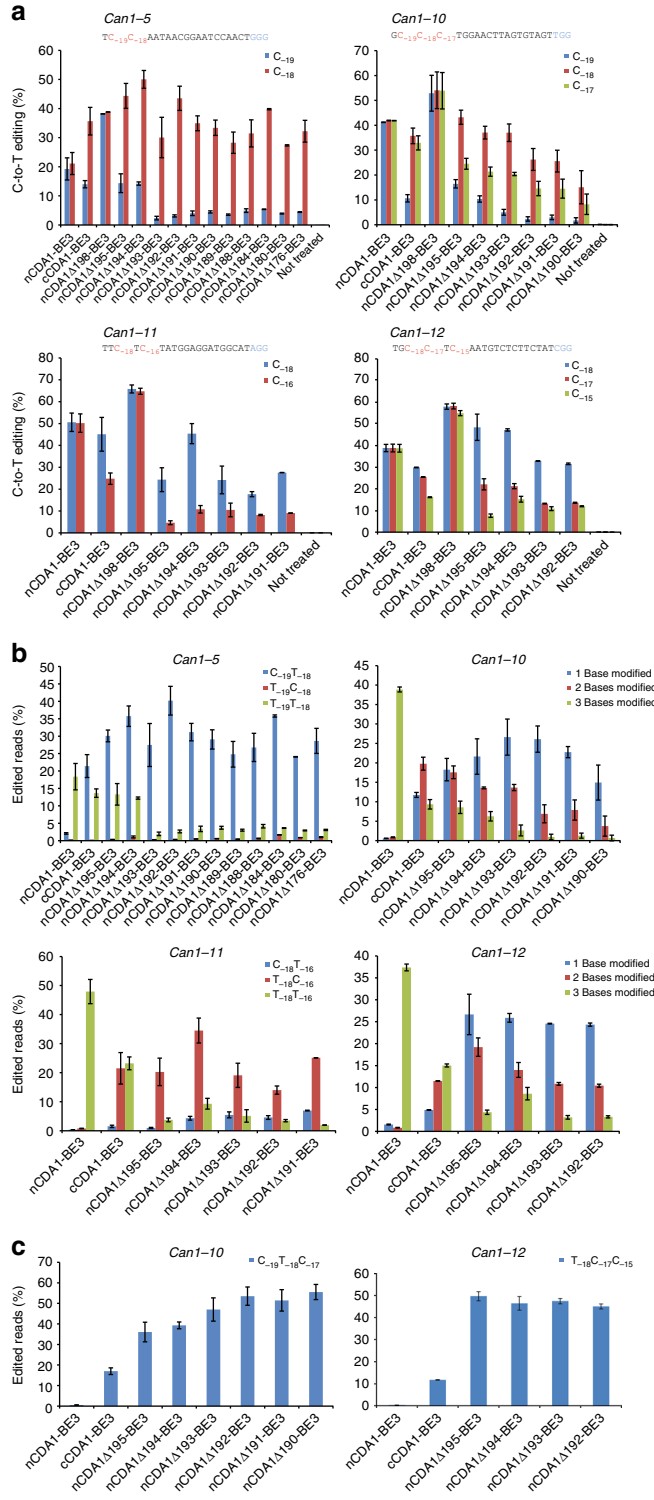

**Fig. 5** Base editors with C-terminally truncated CDA1 domains edit position C$_{-18}$ with high precision. nCDA1-BE3, cCDA1-BE3, and selected base editors with C-terminally truncated CDA1 domains are compared. **a** Editing of genomic loci containing multiple cytidines directly adjacent or in close proximity to C$_{-18}$. Cytidines representing possible editing targets are shown in red with the subscript number representing their position relative to the PAM (blue). **b**, **c** Base editors with truncated CDA1 domains greatly improve editing product distribution and produce predominantly singly C$_{-18}$-modified products. % of edited reads represents the percentage of total sequencing reads containing the products shown. Values and error bars represent the mean and standard deviation of three biological replicates. Source data are provided as a Source Data file

(i.e., in a homozygous fashion), the base editors with truncated CDA1 domains yielded 18–24 colonies that were homozygous for the allele only edited at position C$_{-18}$. Importantly, two of the base editors produced 100% precisely edited homozygous clones (Fig. 6; Table 1).

To exclude the possibility that the exposure time of the target sequence to the base editor affects editing accuracy, we extended the induction time to 40 or 60 h. This did not appreciably affect editing precision, suggesting that the superior performance of our base editors is largely independent of the duration of the exposure of the genome to the base editor (Supplementary Table 2).

## Discussion

In this work, we have developed two alternative strategies to effectively reduce the width of the editing window and, in this way, greatly increase the precision of C-to-T base editors. The use of stiff, proline-rich linkers of specific lengths can significantly narrow the editing window and thus improve the accuracy of editing (Fig. 1). We attribute this to a more narrowly defined distance between the nucleoside deaminase domain and the nCas9 domain of the fusion protein that is likely to result in more precise positioning of the deaminase domain on the target sequence.

At first glance, fusions of CDA1 to the N-terminus of nCas9 seemed to be a poor choice (Fig. 2). However, our finding that large parts of the C-terminus of CDA1 are dispensable for deaminase function offered the possibility to substantially shorten the distance between the deaminase domain and the nCas9 domain. Interestingly, this approach resulted in a substantial gain in editing precision and product purity (Figs. 3–6; Supplementary Figures 10, 11 and 13). It is important to note that, in our study, we used the most stringent assays and the worst-case scenarios in that our target sequences of base editing contained multiple cytidines in close proximity or were even entirely comprised of cytidines ((C)$_8$ or (C)$_9$ motifs). Although our base editors with C-terminally truncated CDA1 domains readily outperformed current base editors, they unsurprisingly, still showed some level of imprecision on these extreme substrates. However, when tested on more normal sequence contexts, our best-performing editors displayed absolute precision in that they (i) produced only correctly edited clones despite the presence of another C directly adjacent to the target C and (ii) edited the target sequence with very high efficiency and accuracy in both alleles of the target gene (Fig. 6; Table 1).

The two types of optimized base editors developed here (BE3-PAPAPAP and base editors with CDA1 truncations) have narrowed activity windows that do not overlap. BE3-PAPAPAP mainly edits within an activity window from −14 to −16, whereas base editors with CDA1 truncations mainly edit at position −18. Consequently, the preference for either BE3-PAPAPAP or a base editor with a CDA1 truncation

We also investigated the indel frequency and base editing purity at these sites when treated by narrowed-window base editors. We found that the frequency of editing errors was very low, consistent with what has been reported for other base editors (Supplementary Figure 13).

Finally, we also determined the base editing outcome in individual colonies obtained by the canavanine selection method. While nCDA1-BE3 and cCDA1-BE3 yielded only 1 and 6 colonies (out of total 24 randomly picked colonies), respectively, that carried the specifically C$_{-18}$ edited *Can1* gene biallelically

largely depends on the distance of the target cytidine from a useable PAM.

Importantly, the strategies described here do not require reduction of the deaminase activity[23,24]. Thus, our narrow-window base editors combine superior editing precision with high editing efficiency and product purity. To increase the genome-targeting scope, engineered Cas9 variants with altered PAM recognition properties (e.g., VQR-Cas9, xCas9 and SpCas9-NG) and naturally occurring Cas9 orthologs (e.g., *Staphylococcus aureus* Cas9 and Cpf1) with different PAM specificities can be employed in combination with our base editors, although specific optimization may be needed to account for differences in three-dimensional structure and/or the positions of N and C termini in phylogenetically more distinct Cas9 variants[31–35].

Highly precise base editors will be essential for future applications of genome editing in gene therapy, site-directed mutagenesis in vivo, and precision breeding. A narrower editing window means fewer target nucleotides. Especially for the correction of disease-causing mutations in gene therapy, the introduction of new mutations in the vicinity of the targeted

nucleotide position is not tolerable[16,22]. We, therefore, expect that our high-precision base editors will find wide applications in many areas of basic and applied research.

## Methods

**Yeast strains and growth conditions**. *Saccharomyces cerevisiae* BY4743 (diploid, *MAT* a/α, *his3Δ1/his3Δ1, leu2Δ0/leu2Δ0, LYS2/lys2Δ0, met15Δ0/MET15, ura3Δ0/ura3Δ*) was used as host strain for genome editing. Cells were grown non-selectively in YPAD medium (2% Bacto peptone, 1% Bacto yeast extract, 2% glucose, 0.003% adenine hemisulfate). For culture in Petri dishes, the medium was solidified with 2% agar. Selection of yeast transformants based on the URA3 and LEU2 markers was done on a synthetic complete (SC) medium (6.7 g/L of Difco Yeast Nitrogen Base, 20 g/L glucose) and a mixture of appropriate amino acids deficient in uracil and leucine (SC-U-L). Yeast strains were cultivated at 28 °C on a rotary shaker.

**DNA methods**. PCR was performed with Phusion High-Fidelity DNA Polymerase (ThermoFisher) according to the manufacturer's instructions. All primers used in this study are listed in Supplementary Table 3. Cloning and amplification of plasmids were carried out in the *E. coli* strain DH5α. Plasmids harboring the *Streptococcus pyogenes cas9* gene (p415-GalL-Cas9-CYC1t) and a chimeric guide RNA construct (p426-SNR52p-gRNA.CAN1.Y-SUP4t) were provided by the laboratory of Dr. George Church and obtained from Addgene (Cambridge, MA, USA). To generate APOBEC1 base editors, the *APOBEC1* reading frame and the partial *cas9* sequence were PCR-amplified using oligonucleotides with overlapping linker sequences (Supplementary Table 3). The two fragments were cloned into the *SpeI/SbfI*-digested p415-GalL-Cas9-CYC1t with the help of the In-Fusion HD Cloning Kit (Clontech, CA, USA). The D10A point mutation was introduced into *cas9* with primers harboring the desired mutation by amplification of the entire plasmid template followed by *DpnI* digestion to remove the parental template. The *UGI* gene was codon-optimized for yeast and synthesized (Eurofins Genomics, Ebersberg, Germany), followed by insertion into the *AscI/MluI*-digested vector p415-GalL-Cas9-CYC1t. To generate CDA1 base editors, the reading frame encoding pmCDA1 was PCR-amplified to replace the *APOBEC1* fragment within BE3, thus generating nCDA1-BE3. To produce a fusion of CDA1 to the C-terminus of Cas9, plasmid pRS315e_pGal-nCas9 (D10A)-PmCDA1 (provided by the laboratory of Akihiko Kondo, Hyogo, Japan, and obtained from Addgene) was modified. First, the amplified *UGI* sequence was introduced into the *XbaI* site, and the resulting vector was then digested with *AscI* and *SphI*. Subsequently, two PCR fragments (overlapping by the XTEN linker sequence) were inserted to generate cCDA1-BE3. Insertion of three PCR fragments (covering XTEN and *APOBEC1*) produced base editor cBE3 (Supplementary Table 3). The CDA1 protein truncations were generated by PCR amplification (Supplementary Table 3), and cloned into *SpeI/SbfI*-digested BE3 or *AscI/SphI*-digested cBE3 vectors to produce the ΔCDA1-Cas and Cas-ΔCDA1 vector series, respectively. To produce YEE-BE3, the mutated APOBEC1 from plasmid pCMV-dCpf1-BE-YEE (provided by the laboratory of Jia Chen, Shanghai, China, and obtained from Addgene) was PCR-amplified and cloned into *SpeI/SbfI*-digested BE3. To generate plasmids expressing sgRNAs that target-specific sites (Supplementary Table 4), the protospacer sequences were introduced by PCR amplification (as part of the primer sequence; see Supplementary Table 3), and the resulting PCR products were cloned into the *ClaI/KpnI*-digested vector p426-SNR52p-gRNA.CAN1.Y-SUP4t with the In-Fusion HD Cloning Kit (Clontech, CA, USA).

**Yeast transformation and genomic DNA extraction**. Yeast cells were transformed with the LiAc/SS carrier DNA/PEG method using 0.5–1 μg plasmid DNA (ref. [36]). Transgenic clones were selected on SC-U-L media and confirmed by PCR analyses (Supplementary Table 3). Yeast genomic DNA was extracted according to a published protocol[37]. PCR products were purified (PCR Purification kit; Macherey-Nagel) and then sequenced.

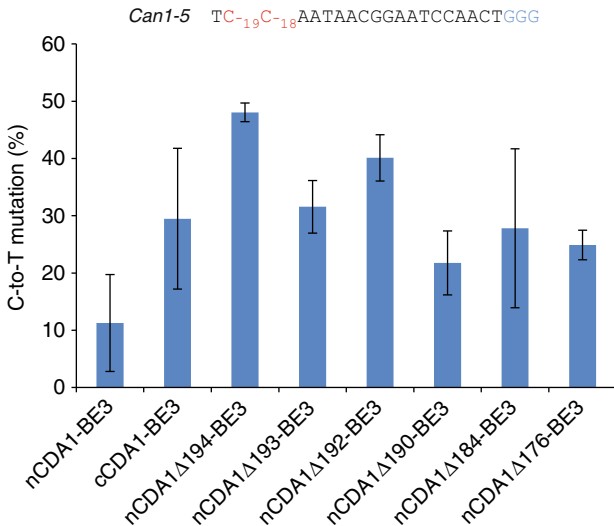

*Can1-5*   TC$_{-19}$C$_{-18}$AATAACGGAATCCAACTGGG

**Fig. 6** Analysis of base editing patterns and efficiencies in single yeast colonies selected for canavanine resistance. A comparison of base editing frequencies for nCDA1-BE3, cCDA1-BE3, and selected base editors with truncated CDA1 domains is shown. Yeast cells were transformed with plasmids expressing the base editor and an sgRNA targeting the *Can1–5* site. The target sequence is shown with the cytidines that can potentially undergo editing in red and the PAM in blue. If C-to-T conversion occurs at position −18 or −19 or both, the *Can1* gene will be inactivated and the cell becomes resistant to canavanine. Values and error bars reflect the mean and standard deviation of three biological replicates. Source data are provided as a Source Data file. See also Table 1

| Table 1 Base editors with CDA1 truncations exhibit many more homozygous C$_{-19}$T$_{-18}$ colonies than nCDA1-BE3 and cCDA1-BE3 | | | | | | | | |
|---|---|---|---|---|---|---|---|---|
| | nCDA1-BE3 | cCDA1-BE3 | nCDA1Δ194-BE3 | nCDA1Δ193-BE3 | nCDA1Δ192-BE3 | nCDA1Δ190-BE3 | nCDA1Δ184-BE3 | nCDA1Δ176-BE3 |
| C$_{-19}$T$_{-18}$ Homozygous | 1/24 | 6/24 | 18/24 | 21/24 | 22/24 | 24/24 | 24/24 | 20/24 |
| C$_{-19}$T$_{-18}$/T$_{-19}$T$_{-18}$ Heterozygous | 0/24 | 11/24 | 2/24 | 2/24 | 1/24 | 0/24 | 0/24 | 2/24 |
| T$_{-19}$T$_{-18}$ Homozygous | 22/24 | 7/24 | 2/24 | 1/24 | 1/24 | 0/24 | 0/24 | 2/24 |

For each base editor, 24 canavanine-resistant colonies were randomly picked from the selection plate followed by sequencing of the *Can1* locus. The major types of edited products are listed in the first column of the table, and the colony numbers representing each product type are given. For nCDA1-BE3, the genotype of the remaining colony is C$_{-19}$T$_{-18}$/T$_{-19}$C$_{-18}$; for nCDA1Δ194-BE3, the remaining two colonies are C$_{-19}$T$_{-18}$/T$_{-19}$C$_{-18}$ and T$_{-19}$T$_{-18}$/T$_{-19}$C$_{-18}$, respectively

**CAN1 mutagenesis**. Yeast colonies were picked, suspended in 3 mL SC medium with 2% glucose and without leucine and uracil, and grown to a stationary phase. The cells were then pelleted, washed twice in sterile water, and then resuspended in SC induction medium with 2% galactose and 1% raffinose, but without leucine and uracil, to an $OD_{600}$ of 0.3. The cells were incubated for 20 h prior to plating on YPAD rich or SC media plates without arginine but with 60 mg/mL ʟ-canavanine (Sigma). After incubation for 3 days, the colony number on each plate was counted. The C-to-T mutation frequency in CAN1 was determined as the ratio of the colony count on canavanine-containing plates to the colony count on YPAD-rich media plates. Each experiment was performed at least three times on different days. To determine the mutation spectrum, colonies were randomly picked and suspended in sterile water, followed by PCR amplification of the relevant CAN1 fragment and DNA sequencing. Control cultures (not treated with base editors) did not produce canavanine-resistant colonies.

**Next-generation sequencing**. Yeast colonies harboring plasmids expressing base editors and sgRNAs were picked from SC-L-U plates, suspended in 3 mL SC-L-U medium with 2% glucose, and grown to a stationary phase. The cultures were then washed twice to remove residual glucose, resuspended in 5 mL SC-L-U medium with 2% galactose and 1% raffinose to an $OD_{600}$ of 0.3, and incubated for 20 h at 28 °C on a rotary shaker. Genomic DNA was extracted from culture samples of 0.5 mL volume, and the regions targeted by base editing were amplified by PCR with primer pairs containing index tags for sample multiplexing (Supplementary Table 3). PCR amplification was performed with the Phusion High-Fidelity DNA Polymerase (ThermoFisher) according to the manufacturer's protocol, followed by product purification with the NucleoSpin Gel and PCR Clean-up kit (Machery-Nagel). The purified index-labeled PCR products were pooled at equal molar ratios. PCR-free library construction and NGS sequencing, demultiplexing by assigning reads to samples, and data filtering (including removal of adaptor sequences, contaminations and low-quality reads from raw reads) were done commercially (BGI, Hong Kong). Sequencing was performed on an Illumina MiSeq 4000 platform in a paired-end way to obtain 150 bp read length for each side and, on average, more than 100,000 reads per sample.

**Data analysis**. The clean FASTQ files obtained after data filtering were further analyzed with python scripts (available at https://github.com/zfcarpe/Cas9Sequencing). Briefly, the "pattern_extract.py" was first applied to scan all sequencing reads and extract the reads with the fixed length of the editing region (and exactly matching the two flanking sequences). This procedure excluded indel-containing and imperfectly matching reads, and allows summarizing each base calling in an alignment-like manner. Subsequent application of the "result_stat.py" script scanned each base within the editing region and calculated the frequency of each base converted to one of the other three bases by dividing the respective read number by the total number of sequencing reads to obtain the percentage of C-to-T editing and the percentage of edited reads with the C converted to any of the other bases. In addition, the script calculates the frequencies of all edited products by scanning each aligned read for conversion of the potential target cytidines. For the analysis of indel frequencies, the sequencing reads were scanned for two exactly matching 10-bp sequences that flank both sides of the region of interest (i.e., the sequence containing the editing sites). Reads without exact matches were excluded from further analysis. By calculating the length of the region, all sequencing reads exactly matching the length of the reference sequence were classified as not containing an indel, otherwise the read was classified as harboring an indel. A shell script "Cas9Sequencing.sh" combined the processes.

**Reporting Summary**. Further information on experimental design is available in the Nature Research Reporting Summary linked to this article.

## Data availability

The data supporting the findings of this study are available within the paper and its supplementary information files. High-throughput sequencing data have been deposited in the National Center for Biotechnology Information Sequence Read Archive database under accession code PRJNA503986 . A reporting summary for this article is available as a Supplementary Information file. The source data underlying Figs. 1, 2b, c, 4a, b, 5 and 6, and Supplementary Figures 2, 4–9, 11–13 are provided as a Source Data file.

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

## Acknowledgements

We thank Dr. Youjun Zhang (MPI-MP) for providing the yeast strain. This research was supported by the Max Planck Society and a grant from the European Union (Horizon 2020, Newcotiana, 760331-2) to R.B.

## Author contributions

R.B., D.K., and J.T. designed the research. J.T. performed most of experiments. F.Z. performed bioinformatic analyses. R.B. and J.T. wrote the manuscript.

## Additional information

**Competing interests:** The authors declare no competing interests.

