## [Peer Review File · Nature Communications]

Reviewers' Comments:

Reviewer #1:

Remarks to the Author:

The authors responded to all substantive comments made by the reviewers as part of the review at Nature journal. They have also added new data showing that their top base editor (BE-PAPAPAP) outperformed a prior base editor with a narrowed editing window (YEE-BE3) at one target as well as data showing that the base window edited by their best-performing base editors was narrow even at long incubation times. The authors notably did not test their base editors in other strains, although the direct comparison between their editors, BE3, and YEE-BE3 is sufficient to argue that the same trends should extend to other organisms in which BE3 and YEE-BE3 have been employed.

I do have a few remaining comments that should be easy to address through small text changes.

1. When testing YEE-BE3, only one target site was tested. I don't think other sites need to be tested; the authors should merely note that the comparisons were limited (e.g. L. 98) and thus cannot be generalized at this time.
2. L. 223, given that the authors do not test any other nucleases, they should at least comment how the different linkers or truncations would extend (e.g. depending on the locations of the N and C termini in phylogenetically distinct Cas9 variants).
3. L. 234 – 239: the authors still need to acknowledge that a narrower window means fewer targets. Also, in the rebuttal, the authors brought up one therapeutic example where their narrower editing window is necessary (APOE4 allele) and mention there are more within the ClinVar database, yet I couldn't find these here or elsewhere in the revised text. My recommendation is to generate an SI figure showing the context of this allele and any more the authors can identify that uniquely require the more compact window.

Reviewer #2:

Remarks to the Author:

The manuscript authored by Tan et al reports that the development of the new high-precision and site-specific base editors in yeast. In the manuscript, the authors first optimized various length linkers between the APOBEC1 and nCas9 in a series of targets, and finally identified that the BE3-PAPAPAP version performed best, which mainly edits within an activity window of -14,-15 and -16. Then the authors developed the base editors using various CDA1 truncations by screening 22 various truncated base editors, which can more precisely edit an activity window of -18. Compared with the previously reported YEE-BE3 and eA3A-BE3, the newly developed narrow-window base editors overcome the limitations of reducing the deaminase activity or depending on the sequence context. I have some minor comments to improve the manuscript.

Minor Comments

1. The authors analyzed the indels and product purity using the new base editors. But they did not detect the off-target effect in the yeast induced by their newly developed BE-PAPAPAP and base editors with CDA1 truncations, just as Nishida et al. (Science, (2016)) did in the yeast using PmCDA1 deaminase. After all, the off-target is a big concern when this system is applied in editing the human genome.
2. In Figure 6, total colonies of each mutation type for nCDA1-BE3 and nCDA1 Δ 194-BE3 were less than 24, what genotypes of the remaining are? The clones were picked from the selection plate, are they false positive or other mutation type? The authors should clarify it in the figures or figure legends.

3. In discussion section, the recently reported SpCas9-NG system should also be discussed, and the related references should be cited.

REVIEWERS' COMMENTS:

Reviewer #1 (Remarks to the Author):

The authors responded to all substantive comments made by the reviewers as part of the review at Nature journal. They have also added new data showing that their top base editor (BE-PAPAPAP) outperformed a prior base editor with a narrowed editing window (YEE-BE3) at one target as well as data showing that the base window edited by their best-performing base editors was narrow even at long incubation times. The authors notably did not test their base editors in other strains, although the direct comparison between their editors, BE3, and YEE-BE3 is sufficient to argue that the same trends should extend to other organisms in which BE3 and YEE-BE3 have been employed.

I do have a few remaining comments that should be easy to address through small text changes.

1. When testing YEE-BE3, only one target site was tested. I don't think other sites need to be tested; the authors should merely note that the comparisons were limited (e.g. L. 98) and thus cannot be generalized at this time.

As suggested, we explain this limitation in the revised ms (p.5)

2. L. 223, given that the authors do not test any other nucleases, they should at least comment how the different linkers or truncations would extend (e.g. depending on the locations of the N and C termini in phylogenetically distinct Cas9 variants).

This is now briefly explained on p.10 of the revised ms.

3. L. 234 – 239: the authors still need to acknowledge that a narrower window means fewer targets. Also, in the rebuttal, the authors brought up one therapeutic example where their narrower editing window is necessary (APOE4 allele) and mention there are more within the ClinVar database, yet I couldn't find these here or elsewhere in the revised text. My recommendation is to generate an SI figure showing the context of this allele and any more the authors can identify that uniquely require the more compact window.

As suggested by the Reviewer, we now specifically state that a narrower editing window means fewer target nucleotides (p.11). Also, we mention two specific therapeutic examples where a narrow window is absolutely necessary: the APOE4 allele and the β -thalassemia allele. We explain this in the Introduction (p. 3), and refer to these examples also in the Discussion.

Reviewer #3 (Remarks to the Author):

The manuscript authored by Tan et al reports that the development of the new high-precision and site-specific base editors in yeast. In the manuscript, the authors first optimized various length linkers between the APOBEC1 and nCas9 in a series of targets, and finally identified that the BE3-PAPAPAP version performed best, which mainly edits within an activity window of -14,-15 and -16. Then the authors developed the base editors using various CDA1 truncations by screening 22 various truncated base editors, which can more precisely edit an activity window of -18. Compared with the

previously reported YEE-BE3 and eA3A-BE3, the newly developed narrow-window base editors overcome the limitations of reducing the deaminase activity or depending on the sequence context. I have some minor comments to improve the manuscript.

Minor Comments

1. The authors analyzed the indels and product purity using the new base editors. But they did not detect the off-target effect in the yeast induced by their newly developed BE-PAPAPAP and base editors with CDA1 truncations, just as Nishida et al. (Science, (2016)) did in the yeast using PmCDA1 deaminase. After all, the off-target is a big concern when this system is applied in editing the human genome.

The objective of our study was to improve window width and product purity. We agree with the Reviewer that off-target effects should be considered when applying genome editing in humans. Our base editors were developed on the basis of BE3 and target-AID (Komor, A. C., et al. Nature, 2016; Nishida, K. et al. Science, 2016) in which the off-target effect were tested. The results were fully consistent with the known specificity of Cas9. Thus, we see no reason to believe that this would be different in our base editors.

2. In Figure 6, total colonies of each mutation type for nCDA1-BE3 and nCDA1 Δ 194-BE3 were less than 24, what genotypes of the remaining are? The clones were picked from the selection plate, are they false positive or other mutation type? The authors should clarify it in the figures or figure legends.

We apologize for this omission. The remaining clones were not false positives, but additional (minor) mutation types. This information has been added to the legend (now Table 1, since Tables within Figures are not allowed).

3. In discussion section, the recently reported SpCas9-NG system should also be discussed, and the related references should be cited.

Done (p. 10).